# Pharmacomicrobiomics in Anticancer Therapies: Why the Gut Microbiota Should Be Pointed Out

**DOI:** 10.3390/genes14010055

**Published:** 2022-12-24

**Authors:** Gabriele Conti, Federica D’Amico, Marco Fabbrini, Patrizia Brigidi, Monica Barone, Silvia Turroni

**Affiliations:** 1Microbiomics Unit, Department of Medical and Surgical Sciences, University of Bologna, 40138 Bologna, Italy; 2Unit of Microbiome Science and Biotechnology, Department of Pharmacy and Biotechnology, University of Bologna, 40126 Bologna, Italy

**Keywords:** gut microbiota, pharmacomicrobiomics, anticancer drugs, multi-omics, tumor-associated bacteria, gut microbiota modulation, microbiome-derived metabolism, tumor microenvironment

## Abstract

Anticancer treatments have shown a variable therapeutic outcome that may be partly attributable to the activity of the gut microbiota on the pathology and/or therapies. In recent years, microbiota–drug interactions have been extensively investigated, but most of the underlying molecular mechanisms still remain unclear. In this review, we discuss the relationship between the gut microbiota and some of the most commonly used drugs in oncological diseases. Different strategies for manipulating the gut microbiota layout (i.e., prebiotics, probiotics, antibiotics, and fecal microbiota transplantation) are then explored in order to optimize clinical outcomes in cancer patients. Anticancer technologies that exploit tumor-associated bacteria to target tumors and biotransform drugs are also briefly discussed. In the field of pharmacomicrobiomics, multi-omics strategies coupled with machine and deep learning are urgently needed to bring to light the interaction among gut microbiota, drugs, and host for the development of truly personalized precision therapies.

## 1. Introduction

Nowadays, research related to the human gut microbiota in health and disease is receiving a *crescendo* of attention, as evidenced by the huge number of scientific papers published in the field (58,920 articles and reviews on PubMed.gov as of 26 September 2022). Indeed, the microbial community that inhabits the gastrointestinal tract is closely connected to human physiology, being fundamental among others for the synthesis of vitamins and the digestion of complex polysaccharides with production of short-chain fatty acids (SCFAs), key metabolites for host homeostasis [1], resistance to colonization by enteropathogens (i.e., the barrier effect) and, not least, the education and modulation of the immune system [2,3]. More recently, the gut microbiota has also been attributed a role in the metabolism of numerous xenobiotics that can enter the human body, from environmental pollutants to therapeutic drugs [4,5,6]. This interaction is bidirectional and multimodal, with xenobiotics being able to promote/inhibit the growth of certain taxa, induce a change in the natural pattern of microbial metabolites and influence virulence, with cascading repercussions on the mutualistic relationship with the host [7,8]. It is therefore not surprising that the gut microbiota is increasingly suggested as a key factor influencing not only the onset and progression of various diseases, but also the response to therapies [9]. In particular, more and more evidence is accumulating in the field of oncology, where the idea is taking hold that there is a more favorable configuration of the gut microbiota associated with enhanced anticancer responses, mitigated side effects, and longer disease-free survival [10,11,12,13]. However, only fragmentary information is currently available, and the individual taxa involved, as well as the underlying mechanisms, are often not known.

Here, we provide a state-of-the-art overview of the direct and indirect role of the gut microbiota in influencing anticancer therapies, as well as patients’ clinical outcomes. In particular, after introducing the concepts of pharmacomicrobiomics and toxicomicrobiomics, we summarize the main chemical modifications of anticancer (pro)drugs (used in targeted and untargeted immuno-chemotherapy) known to be accomplished by the gut microbiota, along with indirect mechanisms of microbiota–drug interactions. Based on this, we discuss microbiome-tailored intervention strategies to improve therapeutic outcomes, involving the use of prebiotics, probiotics, fecal microbiota transplantation (FMT), and antibiotics, as well as innovative biotechnological solutions that use microorganisms or microbial products/processes as a therapeutic target or as a vehicle for the delivery of bioactive molecules or to express certain functionalities. Finally, we emphasize the relevance of multi-omics to unravel microbiota–drug interactions at the mechanistic level, thus enabling the development of machine and deep learning models to predict patients’ outcomes and design personalized precision intervention approaches for better quality of life.

## 2. Pharmacomicrobiomics and Toxicomicrobiomics 

In the multi-omics era, as a result of the Human Genome Project [14] and the subsequent Human Microbiome Project [15], novel branches of pharmacology and toxicology have flourished, namely pharmacomicrobiomics and toxicomicrobiomics with the aim of identifying drug response drivers by elucidating the interactions between drugs and human-associated microbes [4,16]. As anticipated above, a primary role in this context is being attributed to the gut microbiota (Figure 1), which has been shown to influence the absorption, distribution, metabolism, excretion, and toxicity (ADMET) profiles of drugs [17]. In particular, two major microbiota-related phenomena have been proven to affect the therapeutic efficacy of different classes of drugs: (i) bioaccumulation [18], and (ii) microbiome-derived metabolism (MDM) [19,20]. Bioaccumulation occurs when microbes store drugs intracellularly without chemically modifying them. This phenomenon has recently emerged as an important aspect of the host–microbiome–drug relationship, which can lead to altered drug availability and changed bacterial metabolism with potentially very relevant implications for microbiota, pharmacokinetics, side effects, and drug response [18]. Specifically, in 2021, Klünemann et al. [18] described, for the first time, the bioaccumulation of 15 non-antibacterial drugs by 25 gut bacterial strains, characterizing over 70 microbe–drug interactions. In particular, the molecular basis of bioaccumulation was investigated for duloxetine, an antidepressant drug. In this case, chemical, biochemical and metabolomic analysis showed that duloxetine binds to bacterial proteins, including metabolic enzymes, inducing altered metabolism in bioaccumulators; in turn, impaired metabolite secretion created cross-feeding opportunities with marked variations in the overall microbial community. However, the ability of certain taxa to bioaccumulate human-targeted drugs is not well characterized yet, and it has been shown that some drugs (montelukast, roflumilast, etc.) can be bioaccumulated by some microbes and biodegraded by others. As for MDM, the gut microbiota can interact with drugs in many different ways, carrying out partial or complete biochemical transformations, with the yield of more or less active/toxic drug metabolites. It can also indirectly influence the efficacy and toxicity of drugs through the modulation of the host pathways involved in drug metabolism and transport [21,22,23], the competition of microbiome-derived metabolites for the same host targets [23,24], the reactivation of detoxified drug metabolites [25], and the modulation of immune system activity [26,27]. Of course, such interactions strictly depend on the route of administration, with enterally administered drugs having the potential to interact directly with the gut microbiota before being absorbed, while parenterally administered drugs potentially encountering the gut microbial counterpart after hepatic metabolism and biliary excretion, with reactivation of detoxified drugs being the primary concern [9]. More and more studies are investigating the MDM of human-targeted drugs, highlighting how interpersonal microbiota variability is closely linked to interpersonal differences in the drug response [18,20,28,29,30]. For example, screening of 2099 clinical drugs incubated with 76 different gut bacterial strains resulted in the identification of 176 drugs undergoing MDM [20], and another concomitant study reported MDM for 57 drugs covering 28 pharmacological classes [28]. Tested drugs showed varying degrees of metabolization by bacterial enzymes, with conversion to new metabolites or a depletion of their concentration detected in vitro, probably attributable to full conversion to undetectable compounds. In most cases, the high-resolution mass spectrometry (MS) profile of the new metabolites showed small differences to parent drugs and/or a similar MS fragmentation pattern. The most frequently reported structural differences were oxidation (−H_2_; +O), reduction (−O), acetylation and deacetylation (+/−C_2_H_2_O), hydrogenation (+H_2_), hydroxylation (+H_2_O), and propionylation (+C_3_H_4_O). Transformed drugs could exhibit enhanced or reduced toxicity, bioavailability, target affinity, and pharmacological activity, thus a different ADMET profile compared to the parent drug [16,20,28]. No less relevant, gut microbes can impact drug pharmacokinetics by altering the pH through their metabolism, for example through the production of SCFAs [31,32]. Alterations in pH, including those mediated by the gut microbiota, are known to markedly affect drug absorption and bioavailability (through effects on drug dissolution, release, and stability), thereby influencing the drug response and side effects [33,34]. On the other hand, pH also exerts obvious selective pressures on the gut microbiota, inhibiting or, vice versa, promoting the growth of bacteria based on their sensitivity/tolerance to acidic/basic pH [35], thus having (individual-specific) repercussions on microbiota–drug interactions, as described above.

## 3. Gut Microbiome Impact on Immuno-Chemotherapeutics

The above underlines the importance of a systematic mapping of drug–microbiota interactions, to possibly predict and modulate the personal response to drugs. In this section, we focus on immuno-chemotherapeutic agents, a class of pharmaceuticals with strong relevance to patients’ survival but with highly variable clinical outcomes, and summarize evidence of how the gut microbiota can directly or indirectly interact with them. In particular, we deal with untargeted traditional chemotherapy and targeted immuno-chemotherapy, namely monoclonal antibodies and small-molecule inhibitors.

### 3.1. Untargeted Traditional Chemotherapy

Traditional or broad-spectrum chemotherapy has been developed to target cell cycle phases, as cancer cells tend to have a shorter duplication time, making them a preferred target for chemotherapy drugs. Unfortunately, even normal cells can be damaged, thus causing several types of side effects. The gut microbiota has been shown to impact the efficacy and toxicity of some traditional anticancer agents (such as platinum drugs, alkylating agents, anthracyclines, camptothecins, and antimetabolites) through several mechanisms, as detailed below.

Regarding platinum drugs, it should be mentioned that patients responding to oxaliplatin showed elevated serum butyrate levels positively correlated to ID2 (inhibitor DNA-binding 2 protein HLH) and interferon-gamma (IFN-γ) expression by human CD8+ T cells, thus suggesting SCFA-driven promotion of anticancer immunity [36]. Furthermore, it has been shown that *Bifidobacterium bifidum* strains could work synergistically with oxaliplatin to reduce tumor growth, by increasing CD4+, CD8+, effector CD9+ T, and natural killer (NK) cells [37]. 

As for the alkylating agent cyclophosphamide (CTX), it is known to stimulate “pathogenic” TH17 (pTH17) cells through a complex circuitry involving gut microbes and MyD88. In particular, the translocation of Gram-positive bacteria into secondary lymphoid organs in response to CTX could polarize naïve CD4+ T cells towards a TH1 or pTH17 pattern activating bacterial-specific memory T cell responses [38].

Another group of traditional chemotherapeutic drugs are anthracyclines, antitumor antibiotics. Doxorubicin (DOX), a member of this class, has several adverse side effects including damage to the kidney, liver, and gastrointestinal mucosa [39], as well as cardiotoxicity, which limit its clinical doses and application [40,41]. DOX also induces gut microbiota imbalance (i.e., dysbiosis), with possible translocation of microbial components across the compromised intestinal barrier [42]. In particular, it has been shown that lipopolysaccharide (LPS), a molecule of the outer membrane of Gram-negative bacteria, can enter the bloodstream and promote the toll-like receptor 4 (TLR4)-mediated production of a wide range of proinflammatory factors (e.g., tumor necrosis factor-α (TNF-α), interleukin-1 (IL-1), and IL-6), thus contributing to systemic damage [43]. On the other hand, several enterobacteria capable of inactivating DOX, through deglycosylation to 7-deoxydoxorubicinol and 7-deoxydoxorubicinolone, have been identified, including the predominantly environmental species *Raoultella planticola* [44]. Microbial detoxification of DOX could influence its therapeutic concentration in patients, significantly limiting its off-target toxicity.

Gut microbial residents have also been involved in drug reactivation processes and thus in undesirable side effects. Two examples of drugs with increased off-target toxicity due to reactivation by gut microbiota enzymes are irinotecan (IRT) and 5-fluorouracil (5-FU). Irinotecan, a camptothecan analog that blocks DNA replication, is administered for the treatment of pancreatic cancer and colorectal cancer (CRC) [45,46]. IRT is administered in the inactive form, converted to the biologically active form SN38 by hepatic and small intestinal carboxylesterases, and then detoxified in the liver by host UDP-glucuronosyltransferases (in the inactive form SN38-G) before being secreted in the intestine. At the intestinal level, SN38-G can be reactivated by specific microbial enzymes, β-glucuronidases, with severe diarrhea [47]. IRT treatment is also accompanied by an alteration of the gut microbiota composition, with a reduction in health-associated genera such as *Lactobacillus* and *Bifidobacterium* [48], and increased levels of *Clostridium* and *Enterobacteriaceae* taxa including *Escherichia coli*. Such a dysbiotic profile could contribute to toxicity due to mucosal injury/and or inflammation and higher levels of β-glucuronidases in the gastrointestinal tract [49,50]. The drug 5-FU is one of the best studied pyrimidine antagonist agents in CRC therapies, which interferes with thymidylate synthesis, thus inhibiting DNA elongation during DNA replication and repair processes. To improve oral bioavailability, 5-FU can be administered in the form of prodrug such as doxifluridine, which can be converted to 5-FU within cells by pyrimidine phosphorylase. Nonetheless, this prodrug can also be deglycosylated in the active form by microbial thymidine or uridine phosphorylases, resulting in premature intestinal activation and toxicity [28]. Furthermore, bacterial dihydropyrimidine dehydrogenase (e.g., expressed by *E. coli* and *Salmonella enterica*) is able to catalyze the inactivation of 5-FU to 5,6-dihydro-5-fluoruracil in the gastrointestinal tract [51]. Treatment using 5-FU has also been associated with dysbiosis in mouse models, with reduced intra-individual diversity and altered composition [52]. 

Finally, the gut microbiota has been implicated in the detoxification of the folic acid antagonist methotrexate (MTX). The metabolism of MTX to non-toxic 2,4-diamino-N(10)-methylpteroic acid is carried out by the bacterial enzyme carboxypeptidase glutamate 2. This activity has been correlated positively with the relative abundance of *Prevotellaceae* and *Anaeroplasmataceae*, which could help explain the intra-individual variability in therapy efficacy and toxicity [53,54]. At the same time, administration of MTX can alter the gut microbiota profile in a dose-dependent manner, which can lead to changes in immune cell levels and activity [55,56].

### 3.2. Targeted Immuno-Chemotherapy

Targeted immuno-chemotherapeutic agents can be divided into two groups: small-molecule drugs and macromolecules (e.g., monoclonal antibodies, polypeptides, antibody–drug conjugates, and nucleic acids). Here we focus specifically on monoclonal antibodies and small-molecule inhibitors, as they are the main approaches for targeted therapy today. Monoclonal antibodies (mAbs) are high-molecular-weight glycoproteins with high selectivity for their targets, which are typically confined to the cell surface. They are generally administrated intravenously and recognizable thanks to the suffix “-mab” (e.g., bevacizumab, an angiogenesis inhibitor that slows the growth of new blood vessels [57]). On the other hand, small-molecule (<1000 Dalton) inhibitors are able to cross cell membranes and act inside cancer cells, directly promoting cell death, have better patient compliance (having a non-mandatory intravenous route of administration) and are mostly identifiable by the suffix “-ib” (e.g., imatinib, a tyrosine kinase inhibitor used to treat chronic myelogenous leukemia and other different types of cancer [58]).

### 3.3. Monoclonal Antibodies

In the last two decades, advances in biotechnology and molecular biology have led to the development of cancer immunotherapy, a milestone in cancer treatment [59]. In particular, immune-checkpoint blockers (ICBs) have become the forefront of immunotherapy approaches, because of their broad activity in distinct histopathological cancer types and their efficacy against tumor metastasis. The most explored ICBs have two important targets: (i) programmed cell death 1/programmed cell death ligand 1 (PD-1/PD-L1), and (ii) cytotoxic T-lymphocyte-associated protein 4/B7-1 (CD80) or B7-2 (CD86) ligands (CTLA-4/B7-1/B7-2) [60,61,62]. All these proteins are related to T cell inactivation, therefore to a decreased immune activity and reduced anticancer response. Unfortunately, however, less than 30% of patients respond to ICB therapy, showing a heterogeneous outcome. In this context, a large body of evidence is accumulating on the role of the gut microbiota in influencing the success of therapies [63,64].

As regards anti-CTLA-4 therapy, oral administration of *Bacteroides fragilis*, combined with *Burkholderia cepacia* or *Bacteroides thetaiotaomicron,* has been shown to promote a TH1-mediated immune response and intratumoral maturation of dendritic cells in mice, thus favoring a better anticancer immune response [65]. Additionally, FMT from melanoma patients to murine models confirmed that high levels of *B. fragilis* were associated with improved antitumor response. Similar results have been published for PD-1/PD-L1 blockers. In particular, higher microbial diversity and increased relative abundance of *Bifidobacterium*, *Fecalibacterium* and other *Ruminococcaceae* taxa have been correlated with improved mAb efficacy, probably due to increased antigen presentation and improved effector T cell activity in the local tumor microenvironment as well as systemically [66,67]. *Ruminococcaceae*, together with other Clostridiales and *Akkermansia muciniphila* have also been shown to establish “homeostatic” consortia capable of supporting the integrity of the intestinal barrier, thus favoring intestinal and immunological health, fundamental for recovery from cancer [66]. However, it should be noted that a high production of SCFAs, particularly propionate and butyrate (produced by *Ruminococcaceae* members among others), appeared to limit the antitumor activity of anti-CTLA-4, with reduced systemic inflammation and immune activation in tumor-bearing mice [68].

### 3.4. Small-Molecule Inhibitors

Since the approval of the first small-molecule inhibitor (imatinib) in 2001, around 90 targeted antitumor small-molecule drugs have been accepted by the US Food and Drug Administration and the National Medical Products Administration of China [69]. These drugs cover a wide range of target proteins, including kinases, epigenetic regulatory proteins, DNA damage repair enzymes, and proteasomes hitting cancer cells in different metabolic pathways [70,71]. 

As for the gut microbiota role, *Bacteroides ovatus* and *Bacteroides xylanisolvens* have been shown to exert a synergistic activity on tumor size reduction in mice treated with erlotinib, another tyrosine kinase inhibitor used to treat lung cancer [72]. In particular, oral gavage of these gut microbes increased the chemotherapy effect, with a 46% reduction in tumor volume compared to the control. Other tyrosine kinase inhibitors, such as sorafenib (SFN) and regorafenib, can undergo intestinal deglucuronidation, which implies enterohepatic recirculation and consequent improvement in the plasma half-life of such inhibitors [73]. As discussed above for IRT, genes coding for β-glucuronidases are widely present in the gut microbiota [74] and the related MDM event may partly explain the inter-individual pharmacokinetic variability observed for these drugs. Furthermore, the staphylococcal superantigen-like protein 6 enhances SFN sensitivity in hepatocellular carcinoma by inhibiting glycolysis and blocking CD47 signaling [75]. Glycolysis inhibition can be particularly effective against cancer cells with a mitochondrial defect or in hypoxic conditions, and the CD47 pathway can also work as a “don’t eat me” signal to macrophage cells, promoting immune escape in certain types of cancer [76,77].

## 4. Modulation of the Gut Microbiota to Improve the Therapeutic Outcome

Mounting evidence points to the gut microbiota as a promising target for improving therapeutic outcome in oncological diseases [78,79]. To date, several clinical trials have been designed with the aim of investigating the therapeutic potential of manipulating the gut microbiota directly in cancer patients [10,11,13]. Among the modulation strategies, prebiotics, probiotics, and antibiotics are certainly the most widespread historically. In recent years, FMT has also taken on an important role. Below, the state of the art of each microbiota modulation approach is discussed, along with all the clinical trials identified through search on ClinicalTrials.gov (see Figure 2 and Table 1 for clinical trials registered over the past two years and still ongoing). Finally, we discuss bacterial-based anticancer technologies as innovative solutions to directly and specifically modulate the tumor microbiota.

### 4.1. Prebiotics

Prebiotics are defined as “a substrate that is selectively utilized by host microorganisms, conferring a health benefit” [80]. The most used prebiotics include carbohydrates such as galactooligosaccharides (GOS), xylooligosaccharides (XOS), fructooligosaccharides (FOS), fructans, and inulin, as well as other compounds such as polyphenols and polyunsaturated fatty acids (PUFAs). Their beneficial effects are attributable to various mechanisms, including: (i) expansion of beneficial bacteria, (ii) reduction of overt pathogens or pathobionts, and (iii) anti-inflammatory and immunomodulatory activities [79].

In the cancer context, García-Peris et al. [81] conducted a randomized, double-blind, placebo-controlled trial, in which 31 patients with gynecological cancer were given a mixture of fibers (50% inulin and 50% FOS), twice daily from one week before to three weeks after post-surgery radiotherapy (NCT01549782). The prebiotic mixture counteracted the radiotherapy-related drop in *Lactobacillus* and *Bifidobacterium* counts, while reducing tissue damage at the enterocyte level. Many other clinical trials using prebiotics in cancer patients have been designed over the years and some are still ongoing. Among those started in the last two years, it is worth mentioning a clinical trial in the United States, in which researchers aim to evaluate the impact of daily dietary supplementation with a prebiotic based on soluble corn fiber on the gut and tumor-associated microbiota, the immune profile and the therapeutic outcome in 20 patients with stage II and III CRC (NCT05516641). In a Canadian clinical study of 45 patients with advanced non-small cell lung cancer and metastatic melanoma (NCT05303493), researchers will evaluate the safety and tolerability of the Camu-Camu prebiotic in combination with ICBs, as well as the impact on response. Interestingly, the Camu-Camu berry, also known as *Myrciaria dubia*, has recently been shown to lead to the enrichment of *A. muciniphila*, a bacterium associated with favorable clinical outcome in melanoma patients undergoing PD-1 immunotherapy [66], and the improvement of ICB efficacy in preclinical models [82,83]. In the context of a randomized, double-blind, placebo-controlled (maltodextrin) American clinical trial in 30 patients with myeloma or lymphoma undergoing autologous stem cell transplantation (NCT05135351), the authors will evaluate the impact of prebiotic supplementation with resistant starch on the diversity of the gut microbiota at the time of stem cell engraftment. It should be remembered that higher microbiota diversity has been associated with better survival after autologous stem cell transplantation in multiple myeloma and lymphoma [84,85] In particular, the prebiotic intervention will begin 10 days before the infusion of the stem cells and will continue until the first day of engraftment of the neutrophils or for about 30 days in total. The impact on intestinal permeability will also be evaluated. Another American clinical trial in 29 patients undergoing hematopoietic stem cell transplantation will evaluate the benefits of following a diet rich in prebiotics before and during the first 100 days after transplantation, especially in terms of reduction of acute graft-versus-host disease and risk of *Clostridioides difficile* infection (NCT04629430). 

### 4.2. Probiotics

Probiotics are defined as “live microorganisms that, when administered in adequate amounts, confer a health benefit on the host” [86]. Due to the multiple effects on the host, including nutrient metabolism, improved barrier function, alteration of the gut microbiota, direct and/or indirect pathogen antagonism, influence on the gut-brain axis and immunomodulation [87], probiotics are also gaining increasing attention in cancer therapy, and their use has been proposed as non-invasive therapeutic adjuvants or protective agents [88,89]. 

In 2010, the interaction between probiotics, gut microbiota and immune functions in cancer patients undergoing colorectal resection was evaluated for the first time [90]. In a subsequent phase IV randomized clinical trial, the authors observed that daily oral administration of the probiotic yeast *Saccharomyces boulardii* for seven days prior to colorectal resection reduced the levels of pro- and anti-inflammatory cytokines, including IL-10, IL-23A and IL-1β, in the intestinal mucosa and at the same time the incidence of infectious complications (13.3% in patients receiving probiotics vs. 38.8% in the control group) [91]. A few years later, Zaharuddin and colleagues [92] determined the effect of consuming probiotics (*Lactobacillus acidophilus*, *Lactococcus lactis*, *Lactobacillus casei*, *Bifidobacterium longum*, *B. bifidum*, *B. longum* subsp. *infantis*) for 6 months on clinical outcomes and levels of pro-inflammatory cytokines (TNF-α, IFN-γ, IL-6, IL-10, IL-12, IL-17A, IL-17C and IL-22) in 52 patients with CRC. Although chemotherapy-induced diarrhea was observed, CRC patients who received probiotics showed a significant reduction in pro-inflammatory cytokine levels (except IFN- γ) compared to the control group. According to the authors, the combination of probiotic strains could therefore be safely consumed four weeks after surgery in CRC patients, leading to an overall benefit on the intestinal microenvironment and inflammatory profile.

Regarding the most recent clinical trials still in progress, an American pilot study on 40 patients with operable stage I-III breast or lung cancer aims to evaluate the efficacy of administering probiotics before surgery on the gut microbiota and the immune system (NCT04857697). Specifically, patients will receive probiotics orally once on day 1, and then twice or thrice a day for 2–4 weeks prior to standard-of-care surgery. Another American research group is conducting an interventional clinical trial aimed at studying the effect of a new probiotic (marketed by BIOHM^®^) on the breast and gut microbiome (bacteriome and mycobiome), polymicrobial biofilms and quality of life in 50 patients with breast cancer, compared to the placebo group (NCT04362826). Other researchers are evaluating the effect of administering the probiotic strain *Lactobacillus rhamnosus* Probio-M9 in improving response to anti-PD-1 treatment in a clinical trial on 46 patients with liver cancer (NCT05032014). Finally, a team of Korean researchers designed a clinical trial on 40 participants with locally advanced rectal cancer to evaluate the efficacy and safety of total neoadjuvant therapy with *L. lactis* (GEN-001) before surgery (NCT05079503). According to preclinical studies, *L. lactis* could activate CD4+ or CD8+ T and NK cells, and exert synergistic effects with oxaliplatin chemotherapy [93]. 

### 4.3. Antibiotics

The use of antibiotics in cancer patients has several pros and cons. For example, antibiotics have been shown to reduce the size and number of neoplastic lesions [94,95], and help eradicate the colonization of enterotoxigenic *Bacteroides fragilis* in a mouse model of intestinal neoplasia [96]. On the other hand, antibiotic treatment negatively impacts the gut microbiota, reducing biodiversity and contributing to select antibiotic-resistant microorganisms [97]. In this regard, a relevant issue is the co-administration of broad-spectrum antibiotics instead of selective ones against specific pathogens/pathobionts. Targeted antibiotic therapy could help to alter the gut microbiota less while preserving its diversity, which has proven crucial for patients’ response to chemotherapy and prognosis [12,13]. Antibiotic-induced alterations in the gut microbiota could also favorably modulate the pharmacokinetics and efficacy of anticancer drugs. In fact, specific antibiotics could be used to inhibit bacterial species carrying β-glucuronidase activity, thus reducing intestinal toxicity due to unwanted drug reactivation. This is the case of vancomycin, a glycopeptide antibiotic that has been shown to reduce the abundance of β-glucuronidase-expressing bacteria in mouse models, decreasing the gastrointestinal toxicity of several drugs including IRT [98]. Vancomycin could also prove useful in the specific context of 5-FU chemotherapy, since the alterations in the gut microbiota composition induced by its oral administration resulted in a reduced production of microbial-derived dihydropyrimidine dehydrogenase [99], which could counteract the local 5-FU deactivation as discussed above (see section “Untargeted traditional chemotherapy”). 

Over the past two years, several clinical trials have been designed to evaluate the impact of microbiota modulation through antibiotics in cancer patients. In particular, a randomized multicenter study, currently under recruitment, will investigate the impact of early discontinuation of empiric antibiotics for febrile neutropenia in 220 children with cancer, including changes in gut microbiota composition (NCT04637464). In a clinical trial involving 40 participants undergoing mastectomy and implant reconstruction, researchers will determine the effects of post-operative antibiotic treatment on the gut and breast microbiome (NCT05020574). It should be noted that, in the context of mammary neoplasms, the most common tissue expander-related infections are from *Staphylococcus* and *Pseudomonas*, two genera particularly represented in the breast tissue microbiome [100,101]. It can therefore be assumed that patients undergoing mastectomy in the presence of a high abundance of *Staphylococcus* and/or *Pseudomonas* are more likely to develop subsequent infections. In an American phase I study, researchers will evaluate the impact of intravenous administration of cefazolin before surgery on the gut microbiota in 20 patients with stage I-II melanoma (NCT04875728). The immune response and the occurrence of surgical site infections in the three months following surgery will also be assessed. Finally, a single-arm multi-institutional pilot study of 25 participants with surgically resectable pancreatic adenocarcinoma will focus on modulating the gut microbiome to improve the efficacy of neoadjuvant immunotherapy, based on the use of pembrolizumab following chemotherapy (folfirinox) in combination with the antibiotics ciprofloxacin and metronidazole (NCT05462496).

### 4.4. FMT

Consisting of the transfer of fecal material from a healthy donor, FMT is arguably the most direct method of reshaping the gut microbiota. In recent years, FMT has been successfully applied for the treatment of recurrent *C. difficile* infection, achieving a 90% cure rate with acceptable side effects [102]. Several studies have provided evidence of the potential of FMT in treating inflammatory bowel disease, colitis, and digestive system cancer as well [103,104]. With specific reference to cancer, several seminal studies have shown that the microbiota from anti-PD-1 therapy responders, infused by FMT in refractory patients, enhanced tumor-infiltrating T cells and amplified the efficacy of immunotherapy [67,105]. Although large randomized controlled trials are needed, a pilot study has also shown satisfactory results on the feasibility of applying FMT in patients with chronic radiation enteritis [106]. In a randomized clinical trial of 20 patients with metastatic renal cell carcinoma (NCT04040712), Ianiro and colleagues [107] observed resolution of tyrosine kinase inhibitor-induced diarrhea four weeks after FMT with no serious adverse events. 

As for the most recent ongoing clinical trials, an American study on 800 patients with melanoma or genitourinary cancer will evaluate the efficacy of FMT on gastrointestinal complications induced by ICBs (NCT03819296). A second trial on 40 participants will involve washed microbiota transplantation. This new donor fecal sample preparation technique, featuring sequential microfiltration and centrifugation steps, has been shown to be more effective in reducing the rate of adverse events associated with classic FMT treatment, namely fever, diarrhea, abdominal pain, nausea and vomiting [108,109]. FMT safety will also be evaluated on 20 participants with locally advanced or metastatic non-small cell lung cancer undergoing first-line treatment with PD-1/PDL-1 monoclonal antibody (NCT05008861). Researchers will monitor the impact of FMT on gut microbiota dynamics and patients’ immunophenotype. In a clinical trial of 30 participants, researchers aim to demonstrate that FMT from patients responding to ICB therapy can modulate the gut microbiota of patients with ICB-refractory malignant neoplasms, turning them into responders (NCT05273255). Similarly, another phase II study will evaluate the efficacy and safety of FMT in combination with systemic treatment with sintilimab and fruquitinib in 30 patients with CRC refractory to chemotherapy (NCT05279677). In an ongoing prospective clinical study of 80 patients with metastatic lung cancer, FMT will be used in combination with standard chemo-immunotherapy to improve disease control (NCT05502913). Similarly, the safety and efficacy of FMT, as well as the improvement of the response to immunotherapy, will be evaluated in 20 patients with advanced lung cancer (NCT04924374). Finally, in a prospective multicenter randomized phase II clinical trial on 150 participants, the efficacy of FMT in the prevention of allogeneic hematopoietic stem cell transplantation complications, particularly graft-versus-host disease, will be evaluated (NCT04935684). While promising, the application of FMT in cancer therapy is still in its infancy and needs further investigation to reveal the underlying mechanisms. Furthermore, cancer patients often have compromised immune system, making donor selection one of the most critical steps to protect patients from life-threatening infections [110].

## 5. Bacterial-Based Anticancer Technologies

The tumor microenvironment is another ecological niche where microorganisms can proliferate with an extracellular and intracellular localization, in both cancer and immune cells (Figure 3) [111]. As for the gut microbiota, tumor-associated bacteria (TAB), present in primary or distal tumor sites, metastatic lymph nodes or liver metastases, can also influence the efficacy of chemotherapy and immunotherapy with pro- or anti-carcinogenic effects [27,112]. For example, it has recently been found that obligate or facultative anaerobes from *Fusobacteriaceae*, *Enterobacteriaceae*, *Clostridiaceae*, and *Bifidobacteriaceae* families can proliferate in hypoxic niches of solid tumors [111,113]. These localized microbes can alter the therapeutic effect of anticancer drugs through the expression of enzymes that are capable of metabolizing such drugs (e.g., dihydropyrimidine dehydrogenase [51]), by modulating autophagy [114], or enhancing tumor-infiltration and activation of CD8+ T cells [115]. In particular, induction of the autophagy pathway by TAB, such as *Fusobacterium nucleatum*, has been shown to reduce apoptotic cell death, thereby promoting chemoresistance [114]. Nowadays, microbial modulation approaches as described above do not specifically act on the tumor microenvironment, making it necessary to develop targeted intervention strategies [111,116,117,118]. 

Several TAB, such as the facultative anaerobe *Salmonella typhimurium*, are known to actively migrate to the tumor, colonizing hypoxic niches in in vitro studies [119,120]. The chemotaxic ability of these microorganisms could be useful for therapeutic delivery and local bacterial nanofactory technology [121,122]. Different targeted antimicrobial approaches have been developed, allowing for antimicrobials to be delivered to metastatic sites, in order to eliminate unwanted TAB and modulate the local microbial niche. For example, polylactic-co-glycolic acid nanoparticles, coated with the gastric epithelial cell membrane, could be used to deliver clarithromycin to target *Helicobacter pylori*, which is related to a higher risk of gastric cancer development and progression [123]. 

TAB metabolites and/or components could also be an interesting target for cancer treatment. A nanotechnology approach was able to induce LPS-binding fusion protein (LPS-trap) production by cancer cells, useful for limiting the LPS-TLR4 interaction at the tumor site [124]. Specifically, the researchers used a lipid-protamine-DNA nanoparticle gene delivery system to transfect selectively cancer cells, inducing transient expression and secretion of LPS-trap and thus LPS-TLR4 binding block in the tumor microenvironment. This treatment could increase the efficacy of immunotherapy and prevent oncogene activation. Similarly, a triple-layered nanogel DOX-loaded nanoparticle has been developed to be degraded by bacterial lipases, releasing DOX into the tumor, thereby allowing for better tissue specificity, targeted anticancer action and a reduction in side effects [125]. 

Furthermore, genetically modified TAB could be used as external inducible therapeutic agents. These types of bacteria, capable of growing in the tumor microenvironment, could be triggered through external stimuli such as: (i) electromagnetic waves, (ii) magnetic fields, and (iii) ultrasound, to express silent genes and work as a bacterial nanofactory. For example, photosynthetic bacteria or bacteria functionalized with photosensitizer nanoparticles, which can reach and accumulate in tumors, have been designed to generate oxygen under laser irradiation, thereby relieving tumor hypoxia and enhancing reactive oxygen species (ROS) production, with damage to tumor cells and contribution to primary tumor elimination [126,127]. Magnetotactic bacteria, capable of producing particular types of membranous structures, i.e., magnetosomes, have also been evaluated for magnetic hyperthermia cancer therapy. Bacterial magnetosomes could in fact be modified to carry drugs, such as DOX and/or oligotherapeutics, and release them at the cancer site through exposure to alternating external magnetic field [128]. Finally, some anaerobic TAB, which populate the necrotic core of solid tumors, could be engineered to express and secrete exogenous therapeutics molecules, such as anti-CTLA-4 and anti-PD-L1 proteins, under a thermal stimulus. For example, focused ultrasound was used to trigger a temperature-dependent operon that induced the expression of anti-CTLA-4 and anti-PD-L1 by engineered tumor-homing probiotic *E. coli* Nissle 1917 [129]. 

Another innovative biotechnological strategy consists of bacterial-derived antitumor vaccines. Cancer cell membrane vesicles could be used to stimulate immune system activity against related patient tumor cells, and bacterial outer membrane vesicles could be fused with them to enhance cancer-specific immunostimulation. In particular, eukaryotic-procaryotic engineered vesicles could trigger the activity of antigen-presenting cells and subsequent activation of cytotoxic T cells, promoting an enhanced antitumoral immune response. A propidium iodide core, added to the engineered vesicles, could convert near-infrared irradiation into cytotoxic heat, damaging cancer cells and generating supplementary tumor antigen, for a better anticancer response [130].

A final way to target the tumor microenvironment comes from bacteriophages directed against TAB. Due to their target selectivity and infection ability, phages represent very versatile vectors for drug delivery and gene therapy. In this context, some researchers have developed a nanotechnological approach based on bacteriophages that could be useful for eradicating CRC, thanks to a click-chemistry applied to phage and dextran nanoparticles containing chemotherapeutic agents (i.e., IRT) [131]. As observed in mouse models, these functionalized phages provided a multitiered strategy to treat CRC: (i) phage infection of the tumor-promoting TAB *F. nucleatum*, (ii) expansion of SCFA-producing species (e.g., *Clostridium butyricum*) thanks to the prebiotic function of dextran, and (iii) delivery of IRT to eliminate CRC cells in the tumor site.

With continuing technological advances, other more specific, targeted, and inducible cancer therapies that exploit biological bacterial processes are expected to be presented, as a very promising way to overcome actual therapeutic limitations.

**Figure 3 genes-14-00055-f003:**
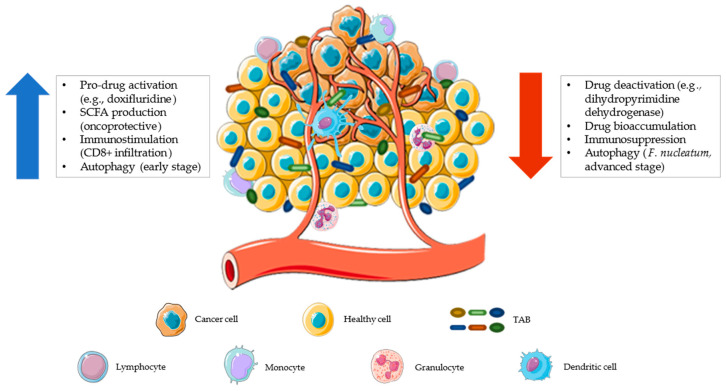
The tumor microenvironment represents a possible ecological niche for facultative and obligate anaerobic bacteria. The increased vascularization, promoted by tumor cells, makes this environment more accessible to leukocytes and bacteria that can interact closely with each other. This interaction can provide anti-tumoral or pro-tumoral activity. Tumor-associated bacteria (TAB) can contribute to anticancer drug resistance by reducing the amount of drugs in the microenvironment (by inactivation or bioaccumulation), exerting immunosuppressive activity, and/or promoting the autophagic pathway that reduces apoptosis [132] (red arrow). Conversely, a tumor microenvironment colonized by health-associated bacteria can contribute to a better therapeutic response by promoting localized pro-drug activation, producing short-chain fatty acids (SCFAs, which help maintain an onco-protective layout), stimulating tumor infiltration by immune cells (CD8+ T lymphocytes), and favoring early-stage autophagy (with antitumor activity [133]) (blue arrow). The figure was partly generated using Servier Medical Art, provided by Servier, licensed under a Creative Commons Attribution 3.0 unported license.

## 6. Multi-Omics Strategy to Study Drug–Microbiota Interactions and Derive Personalized Precision Medicine Models

It is now clear that pharmacomicrobiomics has a concrete relevance in determining the fate and therefore the efficacy of drugs. Studying this phenomenon, however, is more complicated than it may seem as several factors affect the investigation of the relationship between our microbial counterpart and the administered drugs, such as: (i) inter-individual variance and intra-individual fluctuations [134] of commensal microbial ecosystems, as modulated by diet, environment, life habits, immune system and genetics [135,136,137], (ii) the difficulty of deeply characterizing the microbiota composition and function (especially MDM), and (iii) the difficulty of detecting and monitoring the presence of microbial-derived drug metabolites across all body sites. Individualized drug testing before administration may mitigate the first issue, whilst the implementation of several multi-omics techniques such as culturomics, metagenomics, metatranscriptomics and metabolomics might provide stratified information on the compositional and functional structure of the microbiota [138]. What should be taken into account is that the whole human microbial ecosystem (i.e., microbial components, their metabolism and metabolites) changes across body sites. The gut microbiota is certainly the one that has been most studied, but also the oral cavity and our skin (to name a few) represent colonized environments whose effects in health and disease have been ascertained [139,140]. In this scenario, 16S rRNA amplicon sequencing coupled with shotgun metagenomics offer the best tool to date to investigate microbial ecosystem composition down to strain level [141]. Then, the integration of shotgun metagenomics, metatranscriptomics and metabolomics, while challenging, provides a unique opportunity to study the functional potential of the microbiota, from the reconstruction of coding sequences and complete metagenome-assembled genomes to actually transcribed genes with metatranscriptomics, down to the detection of metabolites using untargeted metabolomics, possibly resulting in the identification of genetic determinants of certain MDMs. Untargeted metabolomics using novel high-resolution accurate-mass analytical platforms, for example coupling ultra-high performance liquid chromatography to quadrupole-orbitrap MS, has enabled the identification of novel potential microbial metabolites [142,143] and might help shed some light over microbial drug metabolism. 

A brilliant example of personalized MDM mapping and untargeted multi-omics approach is the work of Javdan et al. [28], who detected novel drug–microbiota interactions and corresponding drug metabolites through liquid chromatography-high resolution tandem MS (HPLC-HRMS/MS), by developing an in vitro MDM screen with patient-derived fecal microbiota. In particular, they isolated metabolizing enzymes by implementing untargeted functional metagenomic screening approaches with comparative transcriptomics, homology-based discovery, and mutagenesis screening, thus paving the way for a personalized platform for the quantification and identification of MDM potential.

Once the determinant(s) of MDM are known, a few pieces are still missing to derive medical recommendations in clinical practice: (i) to estimate the ideal dose of the considered drug, (ii) to evaluate individual additional toxic effects due to each patient MDM, and (iii) to suggest other and/or novel drug classes for treatment, if possible. In this regard, machine learning and deep learning are currently our best tools to detect links between MDM, drug class, administered dose and drug effects, being able to ingest big data of multi-omics platforms and MDM details, possibly recognizing patterns training a model capable of linking such features and deriving the prediction of therapeutic recommendations [144]. The generation of such a model is strictly dependent on the availability of big data concerning MDM and, given that we have just discovered the tip of the iceberg, such powerful informatic tools are probably on a long way to go before effectively contributing to this field. 

An additional in silico tool that might be helpful coupled with untargeted multi-omics approaches is metabolic modelling [145]. Metabolic modelling exploits existing multi-omics data to produce a mathematical modelling approach leveraging genome-scale metabolic model reconstruction of cellular metabolism derived from genomic annotations, deriving biochemical knowledge of biological systems, including metabolite production and metabolite interactions.

## 7. Conclusions

The gut microbiota intertwines a complex and dynamic relationship with the host physiology, exerting, among others, a profound impact on immunological and pathological processes, including the fate and efficacy of drugs. For these reasons, the gut microbiota is increasingly being explored in the precision personalized medicine field. With this review, we wanted to point out the bidirectional interaction between gut microbiota and anticancer pharmaceuticals. Since there are highly subjective clinical responses to the cancer therapies actually in use, understanding this type of interaction could be very important in improving therapeutic outcomes. Pharmacomicrobiomics should therefore become an integral part of the assessment of the ADMET profile of anticancer drugs. However, further studies are needed to unravel the underlying molecular mechanisms and to understand how to target the gut microbiota for better drug efficacy and reduced side effects. In this context, the application of multi-omics strategies is strongly advocated to allow an evaluation as holistic as possible of the composition and function of host microbiomes and to shed light over microbiome-derived drug metabolism. Furthermore, biotechnological approaches, also based on bacteria associated with the tumor microenvironment, should be implemented to provide a more targeted action of the anticancer drug. 

## Figures and Tables

**Figure 1 genes-14-00055-f001:**
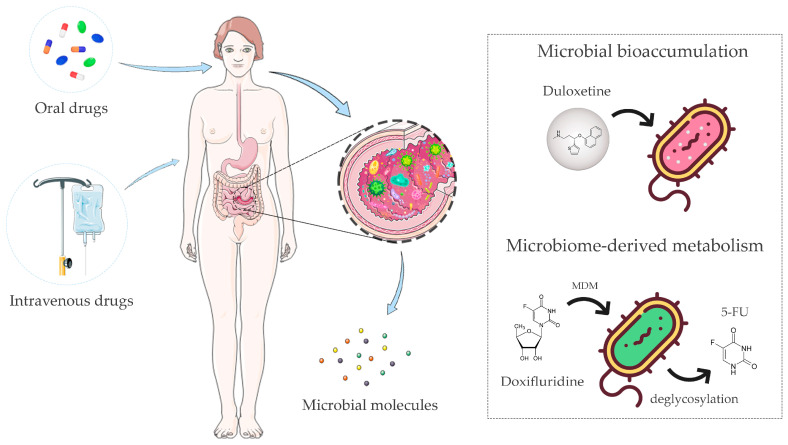
The gut microbiota is able to directly or indirectly interact with drugs, influencing the therapeutic outcome. The gut microbiota has been shown to influence the absorption, distribution, metabolism, excretion, and toxicity profiles of enterally or parenterally administered drugs. These microbiota–drug interactions may result in more or less active/toxic drug metabolites, with a variable impact on the host’s physiology. Microbial bioaccumulation and microbiome-derived metabolism (MDM) are the two main mechanisms involved. Bioaccumulation of duloxetine and MDM of doxifluridine to 5-fluorouracil (5-FU) are shown as examples of these phenomena. The figure was partly generated using Servier Medical Art, provided by Servier, licensed under a Creative Commons Attribution 3.0 unported license and images designed by Freepik.

**Figure 2 genes-14-00055-f002:**
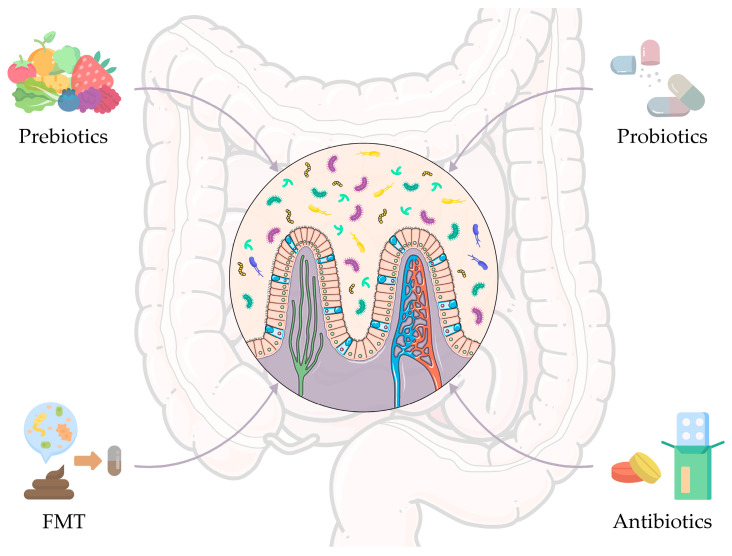
Gut microbiome-based intervention strategies to improve anticancer therapeutic outcomes. Strategies include prebiotics, probiotics, and antibiotics, i.e., the most widespread approaches historically, but also fecal microbiota transplantation (FMT), which has been assuming an important role in recent years. The figure was partly generated using Servier Medical Art, provided by Servier, licensed under a Creative Commons Attribution 3.0 unported license and images from Flaticon resurces.

**Table 1 genes-14-00055-t001:** Clinical trials registered on ClinicalTrials.gov (as accessed in 20 September 2022), started in the last two years and focused on the use of prebiotics, probiotics, fecal microbiota transplantation, or antibiotics for the adjuvant treatment of cancer. Search terms included “gut microbiome”, “prebiotics”, “probiotics”, “fecal microbiota transplantation”, “antibiotics”, and “cancer”. CRC: colorectal cancer; FMT: fecal microbiota transplantation; FOS: fructo-oligosaccharides; GI: gastrointestinal; GOS: galacto-oligosaccharides; HSCT: hematopoietic stem cell transplantation; NSCLC: non-small cell lung carcinoma; PD-1: programmed death-1.

Intervention Type	Title	Status	Study Results	Conditions	Intervention	Location	NCT Number
**Prebiotics**							
	Prebiotics in Rectal Cancer	Recruiting	Not available	Rectal cancer	Soluble corn fiber	United States	NCT05516641
	Camu-Camu Prebiotic and Immune Checkpoint Inhibition in Patients With Non-Small Cell Lung Cancer and Melanoma	Recruiting	Not available	Non-small cell lung carcinoma, melanoma	*Bifidobacterium longum*	Canada	NCT05303493
	Study Using Prebiotics to Improve Gut Microbiome Diversity After Autologous Stem Cell Transplantation	Recruiting	Not available	Multiple myeloma, lymphoma	Resistant starch	United States	NCT05135351
	Effects of Prebiotics on Gut Microbiome in Patients Undergoing HSCT (HCTDiet)	Active, not recruiting	Not available	Multiple myeloma, leukemia, lymphoma	Prebiotic food/drinks (not specified)	United States	NCT04629430
**Probiotics**							
	Effects of Probiotics on the Gut Microbiome and Immune System in Operable Stage I-III Breast or Lung Cancer	Recruiting	Not available	Breast and lung cancer	Probiotics (not specified)	United States	NCT04857697
	Study to Investigate Efficacy of a Novel Probiotic on the Bacteriome and Mycobiome of Breast Cancer	Not yet recruiting	Not available	Breast cancer	BIOHM^®^: *Bifidobacterium breve*, *Saccharomyces boulardii*, *Lactobacillus acidophilus*, *L. rhamnosus*	United States	NCT04362826
	Probiotics Enhance the Treatment of PD-1 Inhibitors in Patients With Liver Cancer	Recruiting	Not available	Liver cancer	Probio-M9: *L. rhamnosus*	China	NCT05032014
	Gut Microbiome and Its Immune Modulation in Locally Advanced Rectal Cancer	Not yet recruiting	Not Available	Rectal cancer	GEN-001: *Lactococcus lactis*	Korea	NCT05079503
**FMT**							
	Gut Microbiota Reconstruction for NSCLC Immunotherapy	Not yet recruiting	Not available	Non-small-cell lung cancer	FMT (capsule)	China	NCT05008861
	Role of Gut Microbiome and Fecal Transplant on Medication-Induced GI Complications in Patients With Cancer	Recruiting	Not available	Melanoma	FMT	United States	NCT03819296
	Fecal Microbiota Transplantation in Patients With Malignancies Not Responding to Immune Checkpoint Inhibitor Therapy	Recruiting	Not available	Cancer	FMT	Switzerland	NCT05273255
	FMT Combined With Immune Checkpoint Inhibitor and TKI in the Treatment of CRC Patients With Advanced Stage	Recruiting	Not available	Colorectal cancer	FMT	China	NCT05279677
	Fecal Microbiota Transplantation With Immune Checkpoint Inhibitors in Lung Cancer	Not yet recruiting	Not available	Lung cancer	FMT, placebo antibiotics, placebo FMT	Israel	NCT05502913
	Microbiota Transplant in Advanced Lung Cancer Treated With Immunotherapy	Recruiting	Not available	Lung cancer	FMT	Spain	NCT04924374
	Washed Microbiota Transplantation for The Treatment of Oncotherapy-Related Intestinal Complications	Recruiting	Not available	Cancer	Washed Microbiota Transplantation	China	NCT04721041
	Faecal Microbiota Transplantation After Allogeneic Stem Cell Transplantation (TMF-Allo)	Not yet recruiting	Not available	Leukemia, lymphoma, myeloma	FMT	France	NCT04935684
**Antibiotics**							
	Modulation of the Gut Microbiome With Pembrolizumab Following Chemotherapy in Resectable Pancreatic Cancer	Not yet recruiting	Not available	Pancreatic cancer	Ciprofloxacin, metronidazole	United States	NCT05462496
	Early Termination of Empirical Antibiotics in Febrile Neutropenia in Children With Cancer	Recruiting	Not available	Cancer	Antibiotics suspension	Denmark	NCT04637464
	Microbiome and Association With Implant Infections	Recruiting	Not available	Breast cancer	Cephalexin	United States	NCT05020574
	The Impact of an Antibiotic (Cefazolin) Before Surgery on the Microbiome in Patients With Stage I-II Melanoma	Recruiting	Not available	Melanoma	Cefazolin	United States	NCT04875728

## Data Availability

Not applicable.

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
