# Peer review of "Pharmacomicrobiomics in Anticancer Therapies: Why the Gut Microbiota Should Be Pointed Out"

_genes, 2022, doi:10.3390/genes14010055_

Round 1

Reviewer 1 Report

In this review, the authors address a current topic which is the involvement of the microbiota in the pharmacokinetics and efficacy of anti-cancer drugs. The review is well written, includes interesting information with a large number of references.

I have a few comments to the authors.

1) I would like the authors to better explain the phenomenon of bioaccumulation.

2) Is there an implication of the gastrointestinal pH on the pharmacokinetics of drugs via the disruption of the microbiota?

3) Can we counteract an adverse effect of the intestinal microbiota by changing the route of administration, whether of the anticancer drug or of a co-administered antibiotic?

4) Regarding antibiotics, I think the manuscript did not focus well on the implication of this drug class on altering the microbiota and thus modulating the pharmacokinetics and efficacy of anticancer drugs. I recommend that the authors better address this point and enrich it with examples.

5) I recommend that the authors give examples of tumor-localized microorganisms and explain their mechanism in modulating the effects of anticancer drugs.

6) My last comment concerns the figures. I feel that both figures are very general. I suggest to the authors to strengthen the review with more illustrative and explanatory figures.

Author Response

Reviewer #1

In this review, the authors address a current topic which is the involvement of the microbiota in the pharmacokinetics and efficacy of anti-cancer drugs. The review is well written, includes interesting information with a large number of references.

I have a few comments to the authors.

1) I would like the authors to better explain the phenomenon of bioaccumulation.

We would like to thank the Reviewer for appreciating our work and apologize for the lack of details on the bioaccumulation phenomenon.

In the revised version of our manuscript, we have added a few lines to the main text, specifying what the phenomenon consists of and discussing the case study of duloxetine (an antidepressant drug) as reported in the original work by Klünemann et al., 2021 (doi: 10.1038/s41586-021-03891-8) (please, see L81-82, 89-94).

2) Is there an implication of the gastrointestinal pH on the pharmacokinetics of drugs via the disruption of the microbiota?

We are grateful to the Reviewer for pointing out this relevant aspect of drug pharmacokinetics.

As suggested, we have added a focus in L126-135, where we discuss the bidirectional relationship between gut pH and microbiota, and its impact on pharmacokinetics. In short, pH exerts obvious selective pressures on the gut microbiota, inhibiting or vice versa promoting the growth of bacteria based on their sensitivity/tolerance to acidic/basic pH (Firrman et al., 2022 doi: 10.1093/femsec/fiac038). pH is therefore a driver of community structure and function, which may have potential (individual-specific) repercussions on pharmacokinetics, as extensively discussed in our manuscript. On the other hand, gut microbes can obviously modify the intestinal pH through their metabolism, for example through the production of short-chain fatty acids (which, as expected, also exert an antibacterial action) [Campbell et al., 1997 doi:10.1093/JN/127.1.130; Den Besten et al., 2013 doi:10.1194/JLR.R036012] Alterations in pH, including those mediated by the gut microbiota, can markedly affect drug absorption and bioavailability (through effects on drug dissolution, stability, release, and stability), thereby influencing drug response and side effects [Abuhelwa et al., 2016 doi:10.1208/S12248-016-9952-8; Stillhart et al., 2020 doi:10.1016/J.EJPS.2020.105280].

3) Can we counteract an adverse effect of the intestinal microbiota by changing the route of administration, whether of the anticancer drug or of a co-administered antibiotic?

Again, we thank the Reviewer for raising this interesting point.

Of course, changing the administration route of drugs can have an impact on the pharmacological profile, also due to the different interactions with gut microbes. In fact, while following enteral administration the drugs can directly interact with the gut microbiota (via the mechanisms discussed in the manuscript) before being absorbed, those administered parenterally may encounter the gut microbial counterpart only after hepatic metabolism and biliary excretion, with the reactivation of detoxified drugs representing the main concern. For what concerns chemotherapeutics, it must be said that the majority of them are administered parenterally, often as an obligate choice deriving from the complex molecular structure, formulation, and pharmacological profile. As regards co-administered antibiotics, in addition to the route of administration, another relevant issue is the use of broad-spectrum antibiotics instead of selective ones against specific pathogens/pathobionts. Targeted antibiotic therapy could help preserve the gut microbiota diversity, which has been shown to be fundamental for patients’ response to chemotherapy and prognosis, or specifically inhibit bacterial species carrying beta-glucuronidase activity, thus reducing intestinal toxicity due to unwanted drug reactivation. This information has been added to the main text (L105-110, L433-444).

4) Regarding antibiotics, I think the manuscript did not focus well on the implication of this drug class on altering the microbiota and thus modulating the pharmacokinetics and efficacy of anticancer drugs. I recommend that the authors better address this point and enrich it with examples.

We apologize to the Reviewer for the lack of proper discussion on antibiotics.

In an attempt to address this point, we have stressed that antibiotics can both negatively and positively modulate the pharmacokinetics and thereby the efficacy of anticancer drugs through induced alterations in the gut microbiota. In particular, we have discussed the example of vancomycin, whose administration during chemotherapy treatments could help control unwanted deactivations of active drugs or reactivations of detoxified ones (L428-449).  

5) I recommend that the authors give examples of tumor-localized microorganisms and explain their mechanism in modulating the effects of anticancer drugs.

We thank the Reviewer for pointing out this gap in the text. As suggested, we have better elucidated the role of tumor-associated bacteria in modulating the effects of anticancer drugs directly at the tumor site. In particular, it is known that several obligate or facultative anaerobes are able to proliferate in hypoxic tumor niches, altering the therapeutic effect of anticancer drugs through mechanisms partly superimposable to those already described for the gut microbiota. Among them, we have briefly discussed the induction of autophagy and its cascading effects on apoptosis and chemoresistance (L528-L537).

6) My last comment concerns the figures. I feel that both figures are very general. I suggest to the authors to strengthen the review with more illustrative and explanatory figures.

We thank the Reviewer for her/his suggestions.

We have modified both Figure 1 and Figure 3 (formerly Figure 2), and their captions accordingly, in order to make them more illustrative and explanatory. Moreover, we have added a new Figure 2 on potential microbiome-based strategies to improve anticancer therapeutic outcomes.

Reviewer 2 Report

The review article ‘Pharmacomicrobiomics in anticancer therapies: why the  gut microbiota should be pointed out’ is a well conceived article.

Overall observation

The abstract gives a comprehensive view of the contents. However, reading the abstract, an impression that, an extensive review is included on nano-biotechnological approaches, which is not present in the text as a separate section. Hence, the sentence may be modified, or review be included

The introduction is scripted meticulously, that the reader is provoked to continue into the article.

The flow of the information in each section is organized into a story line, that, the reader can easily comprehend what the author wants to convey.

The conclusion provides researchers an opportunity to think of future directions to follow in the field of gut microbial research.

However, the authors have included some of the registered studies which may need approval from the concerned researchers.

Author Response

Reviewer #2

The review article ‘Pharmacomicrobiomics in anticancer therapies: why the  gut microbiota should be pointed out’ is a well conceived article.

Overall observation

The abstract gives a comprehensive view of the contents. However, reading the abstract, an impression that, an extensive review is included on nano-biotechnological approaches, which is not present in the text as a separate section. Hence, the sentence may be modified, or review be included

The introduction is scripted meticulously, that the reader is provoked to continue into the article.

The flow of the information in each section is organized into a story line, that, the reader can easily comprehend what the author wants to convey.

The conclusion provides researchers an opportunity to think of future directions to follow in the field of gut microbial research.

However, the authors have included some of the registered studies which may need approval from the concerned researchers.

We thank the Reviewer so much for meticulously reading and appreciating our work.

As suggested, we have modified the part on nano-biotechnological approaches in the Abstract (it was actually misleading, sorry!), and better explained the focus of our review (L18-20).

As for registered studies, we are aware it is a risky choice but dictated by the desire to show the most recent trends in the field of microbiota manipulation. In any case, if the Reviewer deems it appropriate, we may remove the studies who are not yet recruiting.

Round 2

Reviewer 1 Report

I thank the authors for their answers to all my questions. 

I have no further comments for the authors